# Loss of *Nnt* Increases Expression of Oxidative Phosphorylation Complexes in C57BL/6J Hearts

**DOI:** 10.3390/ijms22116101

**Published:** 2021-06-05

**Authors:** Jack L. Williams, Charlotte L. Hall, Eirini Meimaridou, Lou A. Metherell

**Affiliations:** 1Centre for Endocrinology, William Harvey Research Institute, Charterhouse Square, Barts and the London School of Medicine and Dentistry, Queen Mary University of London, London EC1M 6BQ, UK; jack.williams@qmul.ac.uk (J.L.W.); charlotte.hall@qmul.ac.uk (C.L.H.); 2School of Human Sciences, London Metropolitan University, London N7 8DB, UK; e.meimaridou@londonmet.ac.uk

**Keywords:** C57BL/6N, C57BL/6J, NNT, RNAseq, cardiomyopathy, cellular respiration

## Abstract

Nicotinamide nucleotide transhydrogenase (NNT) is a proton pump in the inner mitochondrial membrane that generates reducing equivalents in the form of NAPDH, which can be used for anabolic pathways or to remove reactive oxygen species (ROS). A number of studies have linked NNT dysfunction to cardiomyopathies and increased risk of atherosclerosis; however, biallelic mutations in humans commonly cause a phenotype of adrenal insufficiency, with rare occurrences of cardiac dysfunction and testicular tumours. Here, we compare the transcriptomes of the hearts, adrenals and testes from three mouse models: the C57BL/6N, which expresses NNT; the C57BL/6J, which lacks NNT; and a third mouse, expressing the wild-type NNT sequence on the C57BL/6J background. We saw enrichment of oxidative phosphorylation genes in the C57BL/B6J in the heart and adrenal, possibly indicative of an evolved response in this substrain to loss of *Nnt.* However, differential gene expression was mainly driven by mouse background with some changes seen in all three tissues, perhaps reflecting underlying genetic differences between the C57BL/B6J and -6N substrains.

## 1. Introduction

The heart is an energetically demanding tissue, requiring constant oxygen supply to support the aerobic metabolism needed to fuel myocardial contractions. Mitochondria are the site of oxidative phosphorylation, the mechanism for ATP generation in aerobic conditions and as such are vital to the central function of the heart [1]. However, the role of mitochondria is not solely limited to producing ATP through oxidative phosphorylation. They are also required to regulate reactive oxygen species (ROS) levels, buffer cytosolic calcium ions and mediate apoptosis through the mitochondrial permeability transition pore [1,2]. Nicotinamide nucleotide transhydrogenase (NNT) is a proton pump located in the inner mitochondrial membrane [3]. The forward reaction of this pump catalyses the reduction of NADP to NAPDH coupled with the oxidation of NADH to NAD and the import of one H^+^ ion into the mitochondrial matrix, providing approximately 45% of cellular NADPH [4]. NADPH is utilized as a reducing agent by antioxidant enzymes to replenish pools of reduced glutathione and thioredoxin, which are used as co-factors to convert hydrogen peroxide to water [5]. NAPDH is also involved in anabolic reactions such as biosynthesis of iron-sulphur proteins [6], some of which are involved in the tricarboxylic acid (TCA) cycle, the electron transport chain (ETC) and fatty acid oxidation [4,6]. Given this, it is theoretically possible that NNT could play a key role in regulating the function of metabolically demanding tissues such as the heart.

In fact, a number of studies have linked NNT dysfunction to cardiomyopathies [7,8,9], however, in humans, the primary phenotype associated with loss of NNT is primary adrenal insufficiency (PAI). Our lab first identified NNT as a causative gene for PAI in 2012, and subsequent studies have identified more kindreds with PAI carrying mutations in NNT [8,10,11,12,13,14,15,16]. Two studies found cardiac defects in patients with NNT mutations, while another showed reduced NNT activity in failing hearts compared to healthy donors, resulting in lower NAPDH and GSH:GSSG ratio [7,8,9]. Additionally, two studies identified testicular adrenal rest tumours in patients with *NNT* mutations [8,16].

Historically, murine studies of *Nnt* have employed the closely related substrains C57BL/6N (B6N) and the C57BL6J (B6J) as ‘wildtype’ and ‘knockout’ models, respectively. In 2005 it was discovered the B6J mice harbour a deletion of exons 7 to 11 in *Nnt*, while the C57BL/6N (B6N) does not, expressing the full-length wild-type *Nnt* gene [17,18]. Based on studies between the B6J and B6N mice, NNT expression has been linked to cardiomyopathy, hypertension, atherosclerosis and diabetes [7,19,20,21,22]. Furthermore, groups have found differences in cardiac remodelling between these substrains in response to stressors such as exercise, aortic banding and angiotensin, although there are conflicting data for the latter [21,23,24]. *Nnt* has often been posited as the causative gene for these phenotypes, as it is by far the best known and best studied genomic difference between the two strains. However, next-generation sequencing has identified a raft of other genomic variants between strains, ranging from single nucleotide polymorphisms to large deletions [25,26]. This has led to caution when ascribing the cause of any emergent phenotype to *Nnt*, and consideration of other potential genomic causes for example, our recent finding of a *Mylk3* variant as a possible cause of cardiomyopathy in the B6N [20,21,27].

In this study, we compare the cardiac transcriptome of the C57BL/6N and the C57BL/6J mice. However, to isolate *Nnt* effects from those caused by other variants between the strains, we also examined a third mouse, the C57BL/6J-BAC (hereafter referred to as B6JBAC) that expresses the full-length *Nnt* gene on the C57BL/6J background via a bacterial artificial chromosome (BAC) [18]. In particular, we examined whether there are any signatures of cardiac remodelling, redox stress or mitochondrial dysfunction which correlate with *Nnt* expression. Strikingly, we find very little evidence of a role for *Nnt* in cardiac function. Instead we found increased expression of genes segregating with substrain including many associated with oxidative phosphorylation, the TCA cycle and sugar metabolism in the B6J in the heart and adrenal, as well as increased expression of DNA maintenance genes in all three tissues in the B6J. We similarly investigated the transcriptome in the adrenal gland and testes, two highly metabolically demanding tissues which have phenotypes in humans linked to NNT, finding common genes that were regulated by substrain rather than NNT status.

## 2. Results

### 2.1. Differences in Cardiac Transcriptome between the B6N and B6J Are Independent of NNT Expression

To confirm rescue of NNT expression and localisation in the B6JBAC, we isolated protein from liver tissue in three of the mice from each group and purified different cellular fractions. We found NNT expression was rescued and correctly localised to the mitochondria in the B6JBAC (Figure 1A,B). We subsequently isolated RNA from adrenals, testes and hearts of five 18-month old male mice from each group to compare the transcriptome of each tissue across the three groups. We used HISAT2 to align reads, followed by FeatureCounts and Deseq2 to identify differentially expressed (DE) genes (adjusted *p*-value < 0.05) (Figure 1C).

In the heart tissue, the most striking finding was the lack of significantly differentially expressed genes (DEGs) between B6J and B6JBAC mice, as there were only nine genes with an adjusted *p*-value less than 0.05 in this comparison (Figure 1D). In contrast, the B6N had 398 DEGs when compared to the B6J, and 85 compared to the B6JBAC, indicating a dominant effect of the background substrain on the transcriptome (Appendix A). The DEGs showed no skew towards up- or downregulation in any of the comparisons (Figure 1E–G). Fifty-one DEGs in the B6N vs. B6J and B6N vs. B6JBAC were common to both gene lists (Figure 1H), while only two genes were common to the B6N vs. B6J and B6JBAC vs. B6J, *Cntn2* and *Dlgap1* (Figure 1D). Over-representation analysis (ORA) was used to determine whether the DE gene lists contained statistically more than the expected number of genes from a specific pathway. We found significant (FDR < 0.05) enrichment of genes associated with synaptic transmission and inflammation responses in the B6N vs. B6J group (Figure 2A). However, we found no significant pathway enrichment comparing B6J to B6JBAC, perhaps unsurprising given the short gene list (*n* = 9), or in the B6N vs. B6JBAC comparison despite a higher gene count (*n* = 85). Similarly, no significant pathway enrichment was seen in the list of common differentially expressed genes displayed in Figure 1G (data not shown).

We then incorporated direction and magnitude of change as extra parameters to analyse the DEGs by gene-set enrichment analysis (GSEA, Webgestalt). This highlighted similar pathways to those identified in ORA analysis with significant enrichment of IL-17 signalling pathway in the B6J mice when compared with the B6N (Figure 2B). There were fewer pathways identified as enriched when comparing the B6N and B6JBAC, however there were some commonalities with the B6N vs. B6J comparison (Figure 2C). There was significant enrichment of cytokine-cytokine receptor interaction pathways in the B6N relative to the B6JBAC, which also approached significance in the B6N over the B6J. Comparing the B6J and B6JBAC there were too few genes to perform the analysis.

We subsequently analysed the data by Gene Set Enrichment Analysis (GSEA, Broad Institute) using the entire dataset as input. Comparing B6N and B6J a few gene sets were enriched in the B6N, in particular those associated with metabolism (Figure 2D). There was increased expression of genes associated with oxidative phosphorylation in the B6J, particularly those in complex I and V of ETC with decreased expression of many genes involved in glutamatergic and dopaminergic receptor signalling as part of the ‘Neuroactive ligand receptor interaction’ pathway. However, these pathways were not similarly enriched in the B6N compared to the B6JBAC suggesting they may be NNT-associated. The enriched pathways in the B6N vs. B6JBAC comparison contain many genes involved in quality control systems for proteins and mRNA (Figure 2E). Intriguingly, comparing the B6J to the B6JBAC, we found there was differential enrichment of oxidative phosphorylation, DNA replication and TCA cycle pathways, all of which were similarly enriched in the B6J compared to the B6N (Figure 2F) and driven by the same genes, e.g., *Cox10*, *Cox11*, and *Lhpp*. The subset of genes giving this signal were upregulated in the B6J substrain compared with B6N and downregulated in the B6JBAC compared to B6J, reinforcing the conclusion that they are associated with NNT levels. Finally, we sought to determine whether there was any change in the expression of the genes involved in the removal of reactive oxygen species, the supposed principal function of *Nnt.* We found no significant change by GSEA analysis, nor did we find any individual genes within the gene set which showed a genotypic correlation (data not shown).

### 2.2. Differences in Adrenal and Testicular Transcriptomes Are Mostly Independent of NNT Expression

We also analysed the transcriptome of the adrenal and testes, two other tissues with phenotypes linked to NNT dysfunction. We hypothesised there may be some systemic changes in expression which indicate a fundamental role of *Nnt* or other strain-dependent genes. Like the heart [27], the adrenal has high expression of NNT, and the B6JBAC rescues the expression in the B6J mouse to a similar level to the B6N (Figure 3A). For the adrenal, as in the heart, there appeared to be few significant differences (adjusted *p*-value < 0.05) in gene expression between the B6J and B6JBAC (*n* = 34), while there were considerably more differences when comparing the substrains (B6N vs. B6J = 210 genes, B6N vs. B6JBAC = 458 genes) (Figure 3B–E, Appendix A). There was some overlap in DEGs by genotype, with 53 common strain-dependent DEGs (Appendix A) and 7 *Nnt*-dependent (Appendix A). Despite this, there was remarkably little overlap in over-represented pathways among the gene lists for the differential comparisons, with only ‘Focal adhesion’ significantly over-represented in a strain-dependent manner (Figure 3F). However, GSEA analysis of the DEGs showed a significant enrichment in the two B6J substrains in PI3K-Akt signalling, and highlighted other pathways including ‘Metabolic pathways’, ‘Pathways in cancer’ and ‘Focal adhesion’, although these did not reach significance (Figure 3G,H). GSEA analysis on the whole dataset identified many pathways associated with respiration and sugar metabolism significantly enriched in the B6J substrains, with upregulation of genes from all ETC complexes (Figure 3I,J and Appendix A). We also found differences in expression of some genes involved in aldosterone signalling, such as Angiotensin II receptor type 1b (*Agtr1b*). Interestingly, ‘DNA Replication’ appears to be regulated in an *Nnt*-dependent manner across both the adrenal and heart (Figure 3K).

As with the adrenal, the B6JBAC rescued NNT expression to a level which matches that of the B6N (Figure 4A). In the testes, there were only three DEGs between the B6JBAC and B6J, while there were 278 between B6N and B6JBAC, and 1579 in the B6N vs. B6JBAC (Figure 4B, Appendix A). There was considerable overlap between the strain-dependent comparisons, with almost 75% of DEGs in the B6N vs. B6JBAC comparison, also differentially expressed between the B6N and B6J, again indicating that the substrain is the driver of differential gene expression between these mice (Figure 4B–E, Appendix A). With so few genes in the B6JBAC vs. B6J, ORA and GSEA analysis was not possible on the DEG list. ORA analysis on the B6N vs. B6J DEGs revealed many similar pathways to those identified in the other tissues, including those involved in DNA repair, circadian rhythms and hormonal signalling (Figure 4F), with ‘Endocrine and other-factor regulated calcium absorption’ shared with the B6N vs. B6JBAC. GSEA analysis showed no commonality between the DE gene lists (Figure 4G,H). GSEA analysis of the whole dataset identified an enrichment of transcription factor expression in the B6N with respect to the two B6J mice substrains (Figure 4I,J). We also saw an *Nnt*-dependent enrichment of DNA maintenance pathways in all three tissues (Figure 4K). Unlike in the adrenal and heart, there was no enrichment or altered expression of ETC genes, but similar to those tissues, there were no DEGs involved in ROS removal (data not shown).

## 3. Discussion

All three tissues examined in this study have a high metabolic demand: the heart for muscle contraction, and the adrenal and testes for steroid hormone synthesis. In our previous study we showed that the B6N develops dilated cardiomyopathy (DCM) by 12 months, while both B6J and B6JBAC remain healthy [27]. One might hypothesize this would be reflected by differences in energy demand, as the DCM may be induced by a lower threshold of maximal respiration, or may induce a compensatory upregulation in respiratory pathways. While we did find an increased expression of genes involved in oxidative phosphorylation in the B6J compared to the B6N, many of the same genes were also decreased in the B6JBAC compared to the B6J, indicating this effect may indeed be caused by the change in NNT status. A reverse reaction of NNT was discovered in 2015, whereby, under conditions of extremely high respiration, the Krebs cycle is unable to regenerate sufficient NADH to supply the electron transport chain, and so NNT reverses it’s activity, reducing NAD to NADH at the expense of NAPDH oxidation [20]. It is feasible that the increase in oxidative phosphorylation genes in the B6J is a response to the high metabolic demand of a heart. Conversely, in the adrenal gland, the increased expression of oxidative phosphorylation genes in the B6J vs. B6N was also seen in the B6JBAC vs. B6N. This expression pattern may be due to other variants between the B6N or B6J strains, which are not affected by ‘rescuing’ *Nnt* expression in the B6JBAC. We have identified here a number of genes with a strain-dependent expression pattern in multiple tissues, variants in these genes have not been reported, but this may indicate underlying genetic differences between the substrains akin to *Mylk3* and which affect one or more tissues depending on their tissue-specific expression.

One could also postulate strain-dependent changes in gene expression in the adrenal that might alter blood pressure through the renin–angiotensin–aldosterone axis, which could lead to cardiac remodelling. A seminal phenotyping study comparing multiple mouse strains did find increased blood pressure in the B6J compared to the B6N [25]. Of note, we did see a decrease in the expression of *Agtr1b* in the B6J compared to the other two mice. A reduction in signalling through the renin–angiotensin–aldosterone axis may be a response to heightened blood pressure and could cause changes in the cardiovascular system. However, *Agtr1b* was not reduced in the B6JBAC, nor has the blood pressure been measured in the B6JBAC line and so it is unclear whether this is having an effect on the cardiac phenotype between these mice. Furthermore, we found no difference in the concentration of circulating potassium, sodium, chloride or calcium ions in these three strains [27].

We also found that *Nnt* expression correlates with expression of genes involved in DNA maintenance. This may indicate an increase in ROS-induced DNA damage in these tissues. In our previous study, we found increased lipid peroxidation in the B6J, which is caused by excessive ROS [28]. Despite this, we did not see an upregulation of genes involved in reactive oxygen species detoxification in any of the tissues.

We were surprised to find a very modest effect of *Nnt* status on gene transcription when comparing the B6J and B6JBAC mice. In each tissue, this comparison produced the fewest genes, and there were no common genes across tissues. We found far more DEGs in both the other comparisons, perhaps signifying that there is a far greater role for other background variants in regulating the transcriptome and phenotypes of these mice. In our recent paper, we highlighted one such gene, myosin light chain kinase 3 (*Mylk3*) as probably causative of DCM in the B6N mice. Other genes have similarly been identified as causing differential phenotypes between the two common strains, e.g., *Crb1* and *Cyfip2.* It is vital that the scientific community is aware of the differences between these two common strains, as many are still used interchangeably, and may confound the interpretation of data.

## 4. Materials and Methods

The mouse strains used were C57BL/6NTac originally from Taconic (Taconic Biosciences, New York, NY, USA), C57BL/6J distributed by Charles River (Charles River UK), and C57BL/6J mice carrying a BAC transgene to restore murine *Nnt* (officially C57BL/6J-Tg(RP22-455H18)TG1Rdc/H). All mice were 18-month old males and were no more than 10 generations of mating removed from restock. Animal husbandry, tissue and RNA isolation was performed as detailed in [27].

### 4.1. Western Blot

Tissue fractions were prepared following the protocol described in Dimauro et al. [29]. Samples were loaded in SDS buffer without boiling to prevent NNT aggregation. Membranes were imaged using a LiCor Odyssey Fc and analysed using ImageJ.

### 4.2. Bioinformatic Methods

Primary bioinformatic analysis was completed using the DNAnexus platform. FastQC Reads Quality Control version 3.0.1 (28 October 2020) was used for quality control on reads data. 1 testes sample was removed from the analysis as it failed QC (NT-1). HISAT2 [30] version 1.03 (1 June 2017) was used for read alignment, subread_featureCounts [31] version 0.1.0 (21 March 2019) and DESeq2 [32] (R.3.2-packages-quantification modified 11 December 2017) for differential gene expression analysis. The data was filtered by significance determined by an adjusted *p*-value (p.adj) < 0.05 (Benjamini–Hochberg adjusted *p*-value). Secondary analysis was conducted using the WEB-based GEne SeT AnaLysis Toolkit (WebGestalt) [33] accessed in February 2021 and the GSEA software [34,35] that is a joint project between UC San Diego and Broad Institute. WebGestalt uses Gene Ontology (Daily build accessed on 14 January 2019) and KEGG (Release 88.2, 1 November 2018). Rstudio (www.rstudio.com, accessed March 2021) was used to generate the heatmaps, volcano plots, dotplots and bargraphs using the gplots and ggplot2 packages. All fast q files for this research have been deposited in NCBI Bioproject repository and can be accessed via Bioproject ID PRJNA531833, http://www.ncbi.nlm.nih.gov/bioproject/531833.

## Figures and Tables

**Figure 1 ijms-22-06101-f001:**
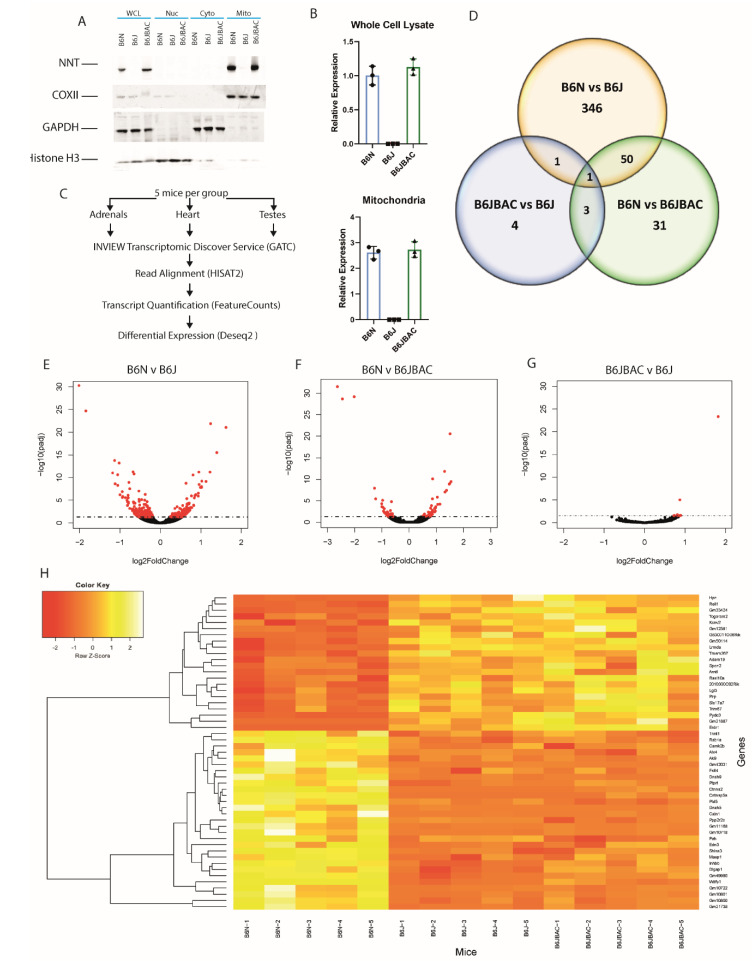
Significant differences in cardiac transcriptome in B6N and B6J are not due to Nnt expression. (**A**) Western blot of NNT expression in liver cellular fractions. WCL—whole cell lysate; Nuc—nuclear fraction; Cyto—cytoplasmic fraction; Mito—mitochondrial fraction; COXII—cytochrome c oxidase subunit II. Representative image of 3 biological repeats. (**B**) Quantification of NNT expression in whole cell lysate and mitochondrial fractins. (**C**) Pipeline of RNA extraction, library preparation and in silico analysis for each tissue. (**D**) Venn diagram showing overlap of DEGs in each pairwise comparison. (**E**–**G**) Volcano plot of gene expression between B6N and B6J (**E**), B6N and B6JBAC (**F**) and B6JBAC vs. B6J (**G**). (**H**) Heatmap of 51 DEGs common to B6N vs. B6J and B6N vs. B6JBAC analyses. *n* = 5 mice per group.

**Figure 2 ijms-22-06101-f002:**
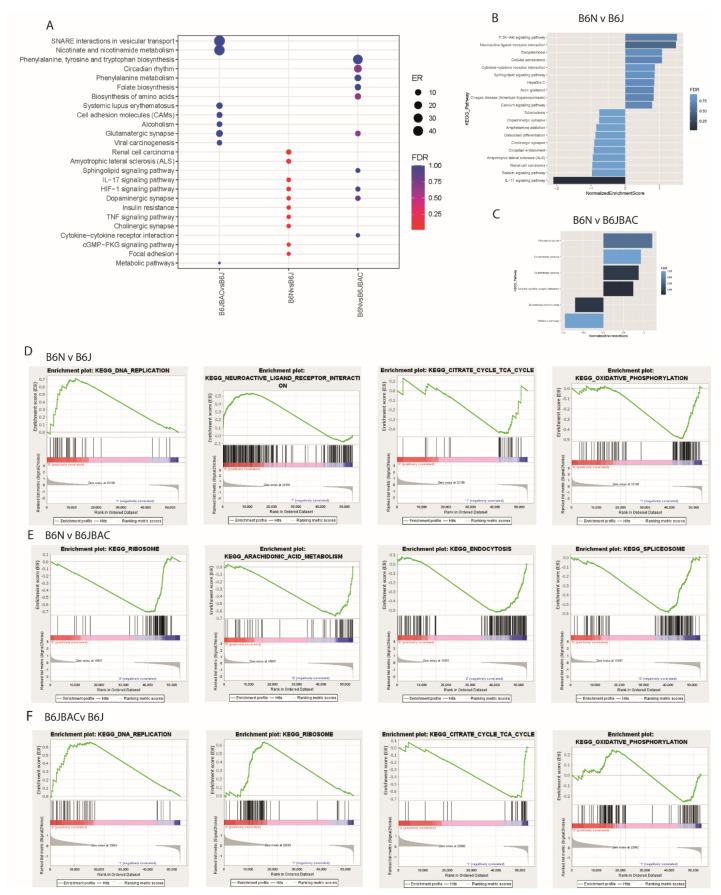
Nnt-dependent expression of pathways associated with respiration in the heart. (**A**) Over-representation analysis (ORA) dot plot of enriched KEGG pathways in DEGs of each comparison. ER—enrichment ratio; FDR—False Discovery Rate. (**B**,**C**) Gene Set Enrichment Analysis (GSEA) of DEGs in the B6N vs. B6J (**B**) and B6N vs. B6JBAC (**C**). (**D**–**F**) GSEA of whole expression dataset for B6N vs. B6J (**D**), B6N vs. B6JBAC (**E**) and B6JBAC vs. B6J (**F**). Selected gene sets displayed with an FDR < 0.25. A negative normalized enrichment score (NES) indicates a relative downregulation of the pathway in the first group compared to the second, and vice versa. *n* = 5 mice per group.

**Figure 3 ijms-22-06101-f003:**
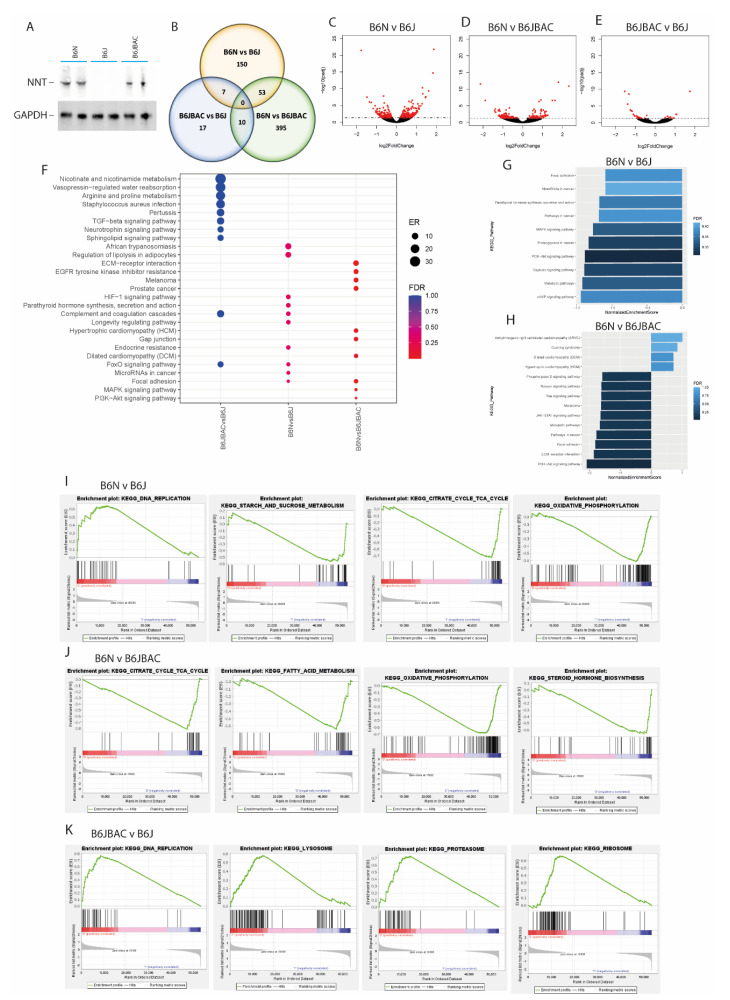
Strain-dependent expression of pathways associated with sugar metabolism and respiration in the adrenals. (**A**) Western blot of NNT expression in adrenal tissue lysates with GAPDH as a loading control. *n* = 2 biological repeats. (**B**) Venn diagram showing overlap of DEGs in each pairwise comparison. (**C**–**E**) Volcano plot of gene expression between B6N and B6J (**C**), B6N and B6JBAC (**D**) and B6JBAC vs. B6J (**E**). (**F**) Over-representation analysis (ORA) dot plot of enriched KEGG pathways in DEGs of each comparison. ER—enrichment ratio; FDR—False Discovery Rate. (**G**,**H**) Gene Set Enrichment Analysis (GSEA) of DEGs in the B6N vs. B6J (**G**) and B6N vs. B6JBAC (**H**). (**I**–**K**) GSEA of whole expression dataset for B6N vs. B6J (**I**), B6N vs. B6JBAC (**J**) and B6JBAC vs. B6J (**K**). Selected gene sets displayed with an FDR < 0.25. A negative normalized enrichment score (NES) indicates a relative downregulation of the pathway in the first group compared to the second, and vice versa. *n* = 5 mice per group.

**Figure 4 ijms-22-06101-f004:**
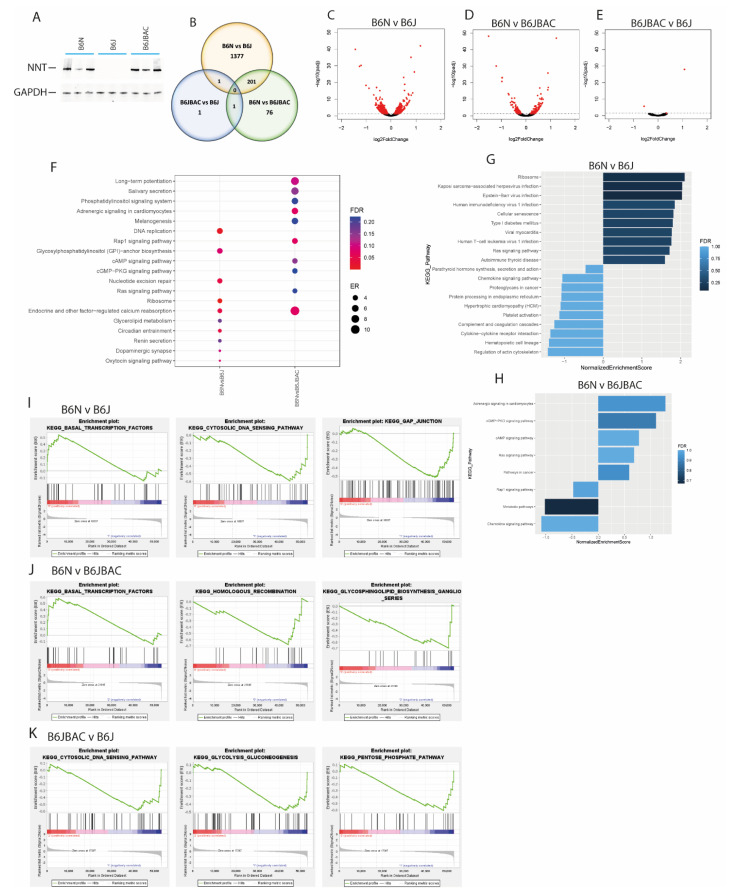
Pathway analysis of testes gene expression. (**A**) Western blot of NNT expression in testes tissue lysates with GAPDH as a loading control. *n* = 3 biological repeats. (**B**) Venn diagram showing overlap of DEGs in each pairwise comparison. (**C**–**E**) Volcano plot of gene expression between B6N and B6J (**C**), B6N and B6JBAC (**D**) and B6JBAC vs. B6J (**E**). (**F**) Over-representation analysis (ORA) dot plot of enriched KEGG pathways in DEGs of each comparison. ER—enrichment ratio; FDR—False Discovery Rate. (**G**,**H**) Gene Set Enrichment Analysis (GSEA) of DEGs in the B6N vs. B6J (**G**) and B6N vs. B6JBAC (**H**). (**I**–**K**) GSEA of whole expression dataset for B6N vs. B6J (**I**), B6N vs. B6JBAC (**J**) and B6JBAC vs. B6J (**K**). Selected gene sets displayed with an FDR < 0.25. A negative normalized enrichment score (NES) indicates a relative downregulation of the pathway in the first group compared to the second, and vice versa. *n* = 5 mice for B6J and B6JBAC, 4 for B6N.

## Data Availability

Data available in a publicly accessible repository. The data presented in this study are openly available in NCBI Bioproject repository, Bioproject ID PRJNA531833. All fast q files for this research have been deposited in NCBI Bioproject repository and can be accessed via Bioproject ID PRJNA531833, http://www.ncbi.nlm.nih.gov/bioproject/531833.

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
