# Peer review of "Loss of Nnt Increases Expression of Oxidative Phosphorylation Complexes in C57BL/6J Hearts"

_ijms, 2021, doi:10.3390/ijms22116101_

Round 1

Reviewer 1 Report

The manuscript  "Loss of Nnt increases expression of oxidative phosphorylation complexes in C57BL/6J hearts" is an interesting study and clearly explored the importance of Nnt in mitochondrial dysfunction and cardiac remodeling.

Minor Comments

  1. What was the age of animals used in this study?
  2. Include western blot of NNT expression in heart cellular fractions.
  3. Figure 1 A quantification data is missing.
  4. In figure-legends, the value of the "n" number is missing.

Reviewer 2 Report

Williams and colleagues offer us an interesting analysis of the transcriptome of the heart, adrenal, and testes from C57BL/6N, C57BL/6J, and C57BL/6J-BAC mice. The goal was to correlate any signatures of cardiac remodeling, redox stress, or mitochondrial dysfunction with Nnt expression. NNT is an important enzyme that regulates mitochondrial NADPH levels and mitochondrial redox balance, which dysfunction is associated with several cardiomyopathies, endothelial dysfunction, etc. The authors claim that, in B6J hearts, loss of Nnt increases the expression of oxidative phosphorylation genes (title); however, the results suggest that those differences are independent of NNT expression. The increased expression of oxidative phosphorylation genes in the B6J strain seems to be mainly guided by other/unidentified variants rather than Nnt. This should be better explained.  

The following points must be corrected before a new revision round:

1- Figure 1 is hiding an important part of the text, at least in the PDF reviewers got access.

2- Gene Set Enrichment Analysis in Figures 2, 3, and 4 are indiscernible.

3- Materials and Methods section should be improved: Mice age; WB information is missing; the “XXXXXXXX” should be replaced.

4- Figure 1: The heatmap represents the “51 DEGs common to B6N v B6J and B6N v B6JBAC analyses”, in the text is enunciated “50 DEGs in the B6N v B6J and B6N v B6JBAC”. This should be clarified. 

5- The authors are surprised with the modest effect of Nnt status on the transcriptome when comparing B6J and B6JBAC, assuming that NNT protein is expressed in the heart, adrenals, and testes of B6JBAC. NNT protein expression is rescued in the Liver of B6JBAC mice; however, NNT protein levels in the other tissues were not analyzed, at least here.  

6- In the discussion, the authors refer to the decrease of Agtrb1b expression in the cardiac tissue of BJ6, but one could not find this mentioned in the results section.

7- There are some typos through the document, e.g. “so NNT reverses it’s activity”

Round 2

Reviewer 2 Report

The authors adequately addressed al my questions and significantly improved their manuscript. It can be published in its present form.